# Oscillatory ERK Signaling and Morphology Determine Heterogeneity of Breast Cancer Cell Chemotaxis via MEK-ERK and p38-MAPK Signaling Pathways

**DOI:** 10.3390/bioengineering10020269

**Published:** 2023-02-18

**Authors:** Kenneth K. Y. Ho, Siddhartha Srivastava, Patrick C. Kinnunen, Krishna Garikipati, Gary D. Luker, Kathryn E. Luker

**Affiliations:** 1Department of Radiology, University of Michigan, Ann Arbor, MI 48109, USA; 2Department of Mechanical Engineering, University of Michigan, Ann Arbor, MI 48109, USA; 3Department of Chemical Engineering, University of Michigan, Ann Arbor, MI 48109, USA; 4Department of Mathematics, University of Michigan, Ann Arbor, MI 48109, USA; 5Michigan Institute for Computational Discovery & Engineering, University of Michigan, Ann Arbor, MI 48109, USA; 6Department of Biomedical Engineering, University of Michigan, Ann Arbor, MI 48109, USA; 7Department of Microbiology and Immunology, University of Michigan, Ann Arbor, MI 48109, USA; 8Biointerfaces Institute, University of Michigan, Ann Arbor, MI 48109, USA

**Keywords:** cancer metastasis, chemotaxis, heterogeneity, single-cell imaging, oscillation, CXCR4 signaling, morphology, data-driven methods, system Identification

## Abstract

Chemotaxis, regulated by oscillatory signals, drives critical processes in cancer metastasis. Crucial chemoattractant molecules in breast cancer, CXCL12 and EGF, drive the activation of ERK and Akt. Regulated by feedback and crosstalk mechanisms, oscillatory signals in ERK and Akt control resultant changes in cell morphology and chemotaxis. While commonly studied at the population scale, metastasis arises from small numbers of cells that successfully disseminate, underscoring the need to analyze processes that cancer cells use to connect oscillatory signaling to chemotaxis at single-cell resolution. Furthermore, little is known about how to successfully target fast-migrating cells to block metastasis. We investigated to what extent oscillatory networks in single cells associate with heterogeneous chemotactic responses and how targeted inhibitors block signaling processes in chemotaxis. We integrated live, single-cell imaging with time-dependent data processing to discover oscillatory signal processes defining heterogeneous chemotactic responses. We identified that short ERK and Akt waves, regulated by MEK-ERK and p38-MAPK signaling pathways, determine the heterogeneous random migration of cancer cells. By comparison, long ERK waves and the morphological changes regulated by MEK-ERK signaling, determine heterogeneous directed motion. This study indicates that treatments against chemotaxis in consider must interrupt oscillatory signaling.

## 1. Introduction

Chemokine CXCL12 and growth factor EGF are important signaling molecules from the microenvironment of a primary breast tumor and distant organs. These molecules contribute to the organ-specific chemotaxis of circulating tumor cells and subsequent metastasis in breast cancer and multiple other malignancies [1,2,3]. Crucial functions of chemotactic signaling through receptor CXCR4 and EGFR in metastatic cancer motivate ongoing efforts to target these signaling pathways with novel therapies [4,5,6]. Studies of these metastatic signaling pathways and compounds targeting them commonly assume that all cells expressing the metastatic signaling receptor signal respond to the corresponding ligand. However, as with other groups, we discovered marked heterogeneity in the activation of intracellular signaling and chemotaxis among seemingly identical cancer cells responsive to multiple kinds of signaling molecules [7,8,9,10,11,12], which complicates the management of metastasis in cancer. Heterogeneous responses of single cells arise from pre-established cell states rather than stochasticity [13,14]. The fact that cells, even in two-dimensional culture, can express a receptor yet fail to respond has critical implications for the development of molecular biomarkers and targeted therapies. The mere presence of a receptor may not correlate with a cancer cell consistently relying on that molecule for oncogenic signaling and functions. Therefore, targeting that receptor might fail to block tumor progression and metastasis. Lack of insights about mechanisms that regulate heterogeneous responses of cancer cells remains a critical barrier as even a single cancer cell that evades therapy could potentially lead to metastatic disease.

Chemotaxis plays an important role in the migration of cancer cells and subsequent metastasis toward chemotactic molecules such as CXCL12 and EGF. Chemotaxis and other cellular functions, such as proliferation, differentiation, and cell death, are regulated by signaling oscillations. Prior research into ERK oscillatory activity has revealed the dynamic nature and integrated feedback loops of signaling networks regulating migration [15,16,17,18,19,20,21,22], proliferation [23], and drug resistance [24]. Crucial functions of oscillatory signals motivate ongoing efforts to understand how they are decoded in cells to drive distinct biological processes. However, how oscillatory signals mediated by chemotactic molecules CXCL12 and EGF can be targeted with novel therapies in cancer remains incompletely understood. Therefore, we aim to understand the processes underlying oscillatory signaling networks that regulate heterogeneous chemotactic responses of cancer cells for the future development of more effective anti-metastasis drugs.

CXCL12–CXCR4 interaction elicits G protein signaling to activate Ras and PI3K and downstream ERK and Akt, which can also be activated by EGF-EGFR. Ras and PI3K drive protrusions of cell membranes, creating positive feedback for Ras activation [19,20]. Ras and PI3K pathways are also restrained by negative feedback and crosstalk mechanisms. mTORC1 functions as a central regulator of signaling with inhibitory effects on both ERK and Akt [25,26]. This interconnected signaling network contributes to oscillatory signals in ERK and Akt and triggers reorganization of the actin cytoskeleton in cells with resultant changes in cell morphology needed for cell migration and chemotaxis [27,28]. Cancer cells with mutations in the Ras-ERK signaling pathway have altered signal transmission properties, such that short pulses of input Ras activity are distorted into abnormally long ERK outputs [29]. Therefore, pre-established cell states will change how cells drive migration through oscillatory signals. In addition, fluctuation of this oscillatory network occurs rapidly on the timescale of minutes after stimulating cells with CXCL12 or EGF, while chemotaxis toward the chemoattractant requires several hours to detect directed movement. Processes cancer cells use to connect oscillatory signaling, morphology, and chemotaxis across these timescales remain incompletely understood, particularly in the context of how heterogeneity in signaling directly relates to heterogeneous chemotaxis among single cells in chemotactic gradient. Furthermore, the inhibitory effects of agents targeting mitogen-activated protein kinases (MAPK), PI3K signaling, and mTORC1 on oscillatory signaling and chemotaxis remain poorly studied. Therefore, in this study, we asked to what extent an oscillatory network of signaling, morphology, and migration in cells associates with heterogeneous chemotactic responses and to what extent targeted compounds impair oscillatory network outputs and chemotaxis.

To answer these questions, we employed an integrated approach combining live, single-cell imaging and quantitative analysis to investigate key signaling pathways and chemotactic responses of MDA-MB-231 breast cancer cells. We implemented multiplexed fluorescence reporters with automated live-cell imaging and image processing to quantify oscillatory information in signaling, morphology, and chemotaxis. Then, we used a time-dependent data processing approach to quantify relationships of signaling, morphology, and chemotaxis within a heterogeneous cell population and across populations treated with targeted inhibitors. We also employed an independent approach termed Variational System Identification (VSI), which combines computation with mathematics to infer the two components of cell motion—random migration and directed motion. Concluding from our independent approaches, we found that amplitudes of short Akt and ERK waves, regulated by MEK-ERK and p38-MAPK signaling pathways, determine the heterogeneous random migration of cancer cells. By comparison, amplitudes of long Akt and ERK waves, the extent of cell elongation, and nuclear polarization determine the heterogeneous directional movement of cancer cells in a chemotactic gradient, which is regulated by MEK-ERK signaling. These data provide new insights into how single cells connect dynamics of signaling response to morphologic changes associated with chemotaxis.

## 2. Material and Methods

### 2.1. Cell Culture and Stable Expression of Fluorescence Reporters in Cells

We purchased MDA-MB-231 human breast cancer cells from ATCC (Manassas, VA, USA) and cultured them in Dulbecco’s Modified Eagle Medium (DMEM) (Thermo Fisher Scientific, Waltham, MA, USA) supplemented with 10% fetal bovine serum (FBS) (Thermo Fisher Scientific), 1% penicillin/streptomycin (Pen/Strep) (Thermo Fisher Scientific), and 1% GlutaMAX (Thermo Fisher Scientific). We maintained the cells at 37 °C in a humidified incubator with 5% CO_2_. Prior to loading cells for chemotaxis experiments, we precultured MDA-MB-231 cells in FluoroBrite imaging media (Thermo Fisher Scientific), 10% FBS, 1% Pen/Strep, and 1% GlutaMAX for 2 days. We used previously engineered cells that have a stable expression of histone 2B fused to mCherry (H2B-mCherry), Akt-KTR (Aquamarine), ERK-KTR (mCitrine), and CXCR4 receptor for our experiments [9].

### 2.2. Chemotaxis Experiments

We used a commercial microfluidic chemotaxis device (µ-Slide chemotaxis, ibidi, Planegg, Germany) with two large reservoirs containing different concentrations of CXCL12 or EGF on both sides of the chambers. We cultured MDA-MB-231 cells in the chemotaxis device in FluoroBrite media (3% FBS, 1% Pen/Strep, and 1% GlutaMAX) for two hours. Depending on the experimental conditions, we preconditioned cells with specific inhibitors for two hours after seeding cells in the device. We used the MEK inhibitor trametinib (Selleck Chemicals, Houston, TX, USA), PI3K inhibitor alpelisib (Selleck Chemicals), and p38 inhibitor SB203580 (Selleck Chemicals).

After preconditioning the cells with an inhibitor, we introduced a chemoattractant gradient with CXCL12 (R&D Systems, Minneapolis, MN, USA) or EGF (R&D Systems). We varied the concentration of CXCL12 on both sides of the device to create chemotaxis in response to different gradients (100 ng/mL to 300 ng/mL versus 0 ng/mL) or chemokinesis (100 ng/mL versus 100 ng/mL in each reservoir, resulting in no effective gradient). For the EGF experiment, we used gradients of 15 ng/mL to 150 ng/mL versus 0 ng/mL. Inhibitors remained in the device for the full 24 h period of chemotaxis.

### 2.3. Automated Time-Lapse Fluorescence Imaging and Image Processing

We took time-lapse multi-color fluorescence images of the chemotactic cells using a fully automated EVOS M7000 imaging system (Thermo Fisher Scientific) at 4 min time intervals for 24 h. The imaging system is equipped with an onstage incubator set to 37 °C, 5% CO_2_, and 80% humidity. The system has autofocusing for live-cell imaging.

After taking time-lapse multi-color fluorescence images, we automatically processed images using custom MATLAB code advanced from our previous version [9,10]. We developed new cytoplasmic segmentation and cell tracking programs to quantify multiplexed data from migrating cells. Briefly, our program segments nuclei and cytoplasm of cells using adaptive thresholding to detect the sharp increase in fluorescence intensity at the edges. The program then conducts a segmented nucleus-assisted watershedding algorithm to segment the merged cytoplasm in each cell. Further, the code extracts the position and morphology properties of each nucleus and cell and quantifies ERK and Akt KTR values as log2 of the cytoplasmic to nuclear KTR fluorescence intensities. The cells are connected between time points using segmented nuclear positions. Our program identifies high-confidence cell matches and other scenarios such as mitosis, missed frames, and merged masks using a statistical approach based on the distance between nuclei and morphological properties. The program trims cell tracks during cell division because signaling and morphology data are not accurate during mitosis.

### 2.4. Single-Cell Analysis

We focus on the heterogeneous responses of individual cells in this study and mainly compare the behaviors of single cells. Different studies with the same conditions are combined to form a collection of single-cell data for each condition. The number of cells n_c_ or the number of aspect ratio peaks n_p_ are reported in each figure legend.

### 2.5. Quantile Regression

We used 20th, 50th, and 80th quantile regression to analyze the linear relationship between two variables [30]. This method estimates the conditional 20th, 50th, and 80th quantiles of the response variable. We used the standard error to calculate the non-zero *p*-value for 50% quantile regression and the 99% confidence interval of the slope and y-intercept.

### 2.6. Earth Mover’s Distance

We used Earth mover’s distance (EMD) to quantify the differences between the single-cell distributions under different experimental conditions. The EMD is a mathematical measure of the distance between two distributions. EMD analyzes two distributions of data as piles of dirt in multivariate space, and the EMD score is the minimum cost of transforming one pile into the other. The higher the EMD score, the greater the difference between the two distributions [31]. We normalized the EMD of the testing condition to the control condition with respect to the EMD of the repeats of the control conditions.
$$Normalized\;EMD=\frac{EMD_{test,control}}{EMD_{control\;1,control\;2}}$$
where $EMD_{A,B}$ is the EMD of condition A from condition B. Note that two controls are different because of biological replicates.

### 2.7. Variational System Identification

We used a data-driven approach called Variational System Identification (VSI) to quantify the overall random migration and directed motion of cells under different conditions. Key to this approach is the treatment of cell populations treated as continuous density fields over appropriately large length scales. The motion of cell populations is the spatio-temporal dynamics of the densities. The advection–diffusion partial differential equation governs these dynamics. The diffusive response in this model represents random migration taken by cells. Advection in this continuous model represents the directed motion of cells, one example of which is chemotaxis in response to chemoattractant. The dominance of advection over diffusion represents a more persistent behavior by the cells. The presence of these mechanisms of motion, as well as the associated parameters in the partial differential equation, are estimated using the techniques of VSI. Details of this method are presented in our previous work with brief details in the Appendix A [32,33,34]. We normalized the random migration and directed motion of drug conditions with respect to the control.
$$Normalized\;random\;migration=\frac{D_{drug}}{D_{control}}$$
Normalized directed motion=v∥,drugv∥,drug2+v⊥,drug2v∥,controlv∥,control2+v⊥,control2
where *D* is the diffusivity inferred from VSI. $v_{∥,A}$ and $v_{⊥,A}$ are the VSI advective velocities parallel to and perpendicular to the gradient, respectively under condition A.

## 3. Results

### 3.1. Amplitudes of Akt and ERK Signaling Waves and Morphological Aspect Ratio Determine Migration Speed of Cancer Cells

CXCL12 signaling through CXCR4 activates kinases Akt and ERK in the PI3K and MAPK pathways, respectively (Figure 1A). Both Akt and ERK promote actin protrusions and cytoskeletal reorganization with resultant changes in cell morphology, cell movement, and chemotaxis [35,36]. We asked to what extent variations in signaling and morphology would contribute to heterogeneous migratory responses in a cell population under the same CXCL12 gradient.

To capture CXCR4 signaling to kinase activities of ERK and Akt, cell morphology, and cell movement in individual chemotaxing cells, we stably expressed fluorescent reporters in cells, cultured them in a microfluidic device, and conducted live-cell imaging over time. CXCR4-expressing breast cancer cells stably expressed histone 2B fused to mCherry (H2B-mCherry) and independent Akt and ERK kinase translocation reporters (KTRs) (Figure 1B). H2B-mCherry marks cell nuclei and enables image segmentation, allowing us to quantify cell chemotactic movement. KTRs translocate reversibly out of the nucleus when the respective kinase is active (Figure 1B). The quantification of the fluorescence intensity ratios in the cytoplasm to nucleus provides analog and independent measurements of Akt and ERK kinase activities in single cells. Further, segmenting the cell based on combined Akt and ERK KTR channels and best-fitting with an ellipse afford quantitative measurements of single-cell morphological properties (Figure 1B). We cultured the fluorescently labelled breast cancer cells in a microfluidic device, which provides a steady chemotactic gradient over time and a favorable environment for long-term migration of cells. Live-cell imaging and automated image processing track cancer cells undergoing migration and mitosis and generate multiplexed single-cell data across time and space. We compared the heterogeneous distributions of Akt, ERK, and cell morphology modulations with the overall migration speed of individual MDA-MB-231 cells exposed to a uniform CXCL12 gradient. This experimental design has the advantage of capturing single-cell multiplexed data across time involving Akt and ERK signaling, cell morphology, and chemotaxis. Additionally, the single-cell multimodal comparison identifies features that modulate migration heterogeneity.

Because Akt and ERK signaling contain both amplitude and frequency information, we decomposed signaling by the range of oscillation period, which is the inverse of frequency, to identify determinant features of signaling in migration (Figure 1C,D). The oscillation period describes the time for one repetitive variation of signaling, while amplitude describes the magnitude of such variation. Complex signal measurements, such as Akt and ERK signals quantified using validated KTRs, are the addition of multiple simple sinusoidal waves with different periods (Figure 1C). To separate Akt and ERK signals into simple waves with different oscillation periods, we applied a bandpass filter that only allows oscillation periods within a certain range. With a fixed range, we varied the oscillation period to identify an optimal period that relates to migration. Signals with various oscillation periods are formed by feedback loops in the signaling network and bolus CXCL12 stimulation. We quantified the overall amplitudes of the filtered waves to compare with the average migration speed in individual cells (Figure 1D). This decomposition method identifies migration-related signaling waves that are hidden by other non-related signaling waves. Additionally, this method detects both gradual changes and pulsing signals to compare broadly among individual cells. Furthermore, since physical changes in the overall cell morphology directly relate to the movement of cells, we used a morphological aspect ratio calculated from the best-fit ellipse to quantify the degree to which cells elongate versus reside in a more circular shape (Figure 1D). We quantified the extent of morphologic changes in cells using the amplitude of aspect ratio changes to compare with the average migration speed in individual cells. This method detects migration-related cell morphology changes that were usually mixed with other cytoskeletal fluctuations.

Heterogeneity comparisons among signaling, morphology, and migration in individual cells show that the amplitudes of signaling and morphology increase with migration speed (Figure 1E). Quantile regressions between the 20th and 80th percentile illustrate that the heterogeneous migration speeds among individual cells exhibit a linear relationship with the amplitudes of signaling and morphology, such as the amplitude of short Akt and ERK waves, long Akt and ERK waves, and morphological aspect ratios. We chose a period range of 28–60 min for short Akt and ERK waves because that is the range of ERK pulses found in previous studies [19,20,21,22], whereas the selected period range of 92–124 min for long Akt and ERK waves matches the period range of morphological aspect ratios. Each aspect of signaling and morphology amplitudes demonstrated a significantly nonzero quantile linear regression (*p* < 0.0001, 99% confidence intervals shown in Appendix A). To determine the extent to which high amplitudes of signaling or morphological change correspond to high migration speeds, we quantified the proportion of top migrating cells that are also top signaling or morphological change cells. We found that more than 55% of top migrating cells have high short ERK wave amplitudes, whereas between 40 and 45% of cells have higher aspect ratio amplitudes, short Akt wave amplitudes, and long Akt and ERK wave amplitudes (Figure 1F). Apart from the above-selected period ranges, the linear relationship between signaling amplitude and speed is also observed in other oscillation periods. The decomposition of Akt and ERK signaling into simple waves of varying periods shows that the linear correlation between Akt and ERK amplitude and migration speed improves when signaling waves become shorter, peaking around 30–40 min (Figure 1G). Despite the close correlation between Akt and ERK amplitudes regardless of oscillation period, ERK amplitudes consistently have a tighter correlation with speed relative to Akt amplitudes (Figure 1F,G). These observations led us to two conclusions: First, Akt and ERK signaling amplitudes, with oscillation period as the regulator, determine how fast cancer cells migrate. The amplitude of ERK signaling waves might be an immediate regulator of migration, while Akt waves relate to migration through ERK. Second, the extent of cell morphological changes also determines the individual migration speed of cancer cells. Long Akt and ERK waves might also relate to the morphological aspect ratio as they share similar ranges of oscillation periods and relationships with migration speed. Our results suggest that cells do not decode the instantaneous kinase activities to drive migration. Rather, cells monitor the rate of change of kinase activities to drive morphological change and migration.

### 3.2. Long ERK Signaling Waves and Aspect Ratios Regulate Heterogeneity of Directed Motion, While Short ERK Signaling Waves Modulate Heterogeneity of Random Migration in CXCR4-Mediated Chemotaxis

Successful chemotaxis is a result of combining overall motive force and the directed movement of cells. Directed movement depends on the chemotactic gradient, while the overall motive force of the cell does not. We next asked to what extent Akt, ERK, and morphology amplitudes depend on the chemotactic gradient and regulate the directed movement of cells (Figure 2A). We conducted experiments with MDA-MB-231 cells exposed to CXCL12 with various levels of gradient (10 ng/mL to 0 ng/mL, 30 ng/mL to 0 ng/mL, 100 ng/mL to 0 ng/mL, and 300 ng/mL to 0 ng/mL) and no gradient (100 ng/mL to 100 ng/mL). Cells in the CXCL12 gradient environment move with directionality, while cells in no CXCL12 gradient environment move randomly (Figure 2B). This analysis method determines whether Akt and ERK signaling and morphology amplitudes regulate single-cell overall motive force or single-cell directed movement due to a chemokine gradient.

To measure the extent of dependence on a CXCL12 gradient, we compared the distribution of signaling or morphology amplitudes under conditions with and without a CXCL12 gradient. The cumulative distributions illustrate that cells exposed to various levels of CXCL12 gradient have a similar distribution of Akt and ERK waves and aspect ratio amplitudes (Figure 2C). In comparison, cells exposed to no CXCL12 gradient show reduced amplitudes of long ERK waves and aspect ratios. We used earth mover’s distance (EMD) to quantify the differences among conditions with the CXCL12 gradient and between conditions with and without a CXCL12 gradient (Figure 2D). By normalizing the differences between conditions with and without gradient with differences among conditions with a gradient, amplitudes of long Akt and ERK waves and aspect ratios have at least a normalized two-fold change, while amplitudes of short Akt and ERK waves do not have much increase in differences between with and without gradient (Figure 2E). These data support that amplitudes of long Akt and ERK waves and aspect ratios depend on a gradient, while amplitudes of short Akt and ERK waves do not depend on a gradient. Since the amplitudes of long Akt and ERK waves and aspect ratios depend on a CXCL12 gradient, we next evaluated the extent that their linear relationship with the migration speed of individual cells holds in an environment with no CXCL12 gradient. The quantile regressions between the 20th and 80th percentile illustrate that the heterogeneous migration speeds among individual cells had poor linear relationships with the amplitudes of long Akt and ERK waves and aspect ratios, respectively (Figure 2F). These parameters no longer demonstrated a significantly nonzero quantile linear regression (99% confidence intervals shown in Appendix A). Concurrent with the property of independence from a CXCL12 gradient, amplitudes of short Akt and ERK waves maintained a linear relationship with migration speed in a no CXCL12 gradient environment (Appendix A). These observations led us to conclude that amplitudes of long Akt and ERK waves and aspect ratios determine the heterogeneity of directed movement in the presence of a CXCL12 gradient, whereas the amplitudes of short Akt and ERK waves determine the heterogeneity of the overall motive force in cancer cells. These results suggest that cells may distinguish between the changes in kinase activities at different rates to drive directed movement and random migration, respectively.

Since short Akt and ERK waves do not depend on a gradient, we next asked if these waves depend on the type of chemoattractant. We exposed MDA-MB-231 cells to gradients of EGF, another driver of cell migration in breast cancer metastasis (Figure 3A) [3]. EGF ligands bind with the receptor tyrosine kinase EGFR, a different class of receptor than the G-protein coupled receptor CXCR4. Comparing CXCL12 to EGF allowed us to determine to what extent short Akt and ERK waves occur and produce similar effects with chemotaxis mediated by two different types of receptors. To measure the extent of dependence on chemoattractant type, we compared the distribution of signaling amplitudes under conditions with an EGF versus CXCL12 gradient. Cumulative distributions illustrate that cells exposed to EGF gradient have larger short Akt and ERK amplitudes (Figure 3B). We also used normalized EMD to quantify differences between EGF and CXCL12 gradients compared to differences among CXCL12 gradients. The amplitude of short Akt waves has a more than three-fold change, while the amplitude of short ERK waves has a less than two-fold change (Figure 3C). These results demonstrate that short Akt wave amplitude changes more than short ERK wave amplitude with EGF relative to CXCL12 as the chemoattractant. As the Akt amplitude depends on the type of chemoattractant, we asked if Akt and ERK signaling amplitudes still maintained a linear relationship with speed with EGF as the chemoattractant. Quantile regressions between the 20th and 80th percentile illustrate that the heterogeneous migration speed among individual cells maintains a linear relationship with amplitudes of short Akt and ERK waves, respectively (Figure 3D). They demonstrate a significantly nonzero quantile linear regression (*p* < 0.0001, 99% confidence intervals shown in Appendix A). However, short ERK wave amplitudes no longer show a tight linear relationship with speed in comparison with short Akt wave amplitude, as evidenced by a similar proportion of top migrating cells having high short Akt and ERK wave amplitudes (Figure 3E) and similar correlation coefficients of Akt and ERK amplitudes with migration speed (Figure 3F) in conditions with an EGF gradient. Together, these observations lead us to conclude that short Akt waves and short ERK waves have distinct roles in determining the heterogeneity of the overall motive force in different classes of receptor-mediated chemotaxis. In CXCR4-mediated chemotaxis, short Akt waves might relate to overall motive force through short ERK waves. However, in chemotaxis driven by EGF, both short Akt and ERK waves contribute to the overall motive force of cancer cells.

### 3.3. Cell Directed Motion Relates to the Extent of Long ERK Wave Reduction, Cell Elongation, and Nuclear Polarization

Because our results showed that long signaling waves and aspect ratios regulate the heterogeneity of directed motion, we next asked to what extent long signaling waves are related to the aspect ratio of cells during migration. When polarized cells reorientate and modulate their morphology during mesenchymal cell migration, nuclear reorientation is important in promoting directed motion [37]. Since ERK and Akt are closely correlated and we have shown that ERK signaling waves directly correlate with migration, we focused on identifying the relationship between long ERK waves and morphology changes during cell migration and quantifying to what extent nuclear polarization impacted ERK signaling, cell morphological change, and migration speed.

We isolated each aspect ratio peak of individual cells and compared them with the corresponding ERK signaling. All aspect ratio peaks were aligned to the same time point to facilitate the identification of time-dependent relationships between aspect ratio and long ERK signaling waves (Figure 4A). Comparing each aspect ratio peak in each cell has the advantage of not missing ERK and aspect ratio temporal relationships from averaging. We found that long ERK waves and morphology changes of a cell interconnect closely (Figure 4B). A majority of aspect ratio peaks aligned with the local ERK minima with little time difference. The amplitudes of each peak pair also correlate. Cells having a large aspect ratio difference at that fluctuation also have a large, negative ERK difference (Figure 4C). They have a significantly nonzero quantile linear regression (*p* < 0.0001, 99% confidence intervals shown in Appendix A). These observations demonstrate that the long ERK waves and morphological shapes of the cells are closely related with a negative correlation. It suggests that larger ERK drops produce greater elongation of cells, potentially to sense a gradient of CXCL12.

To quantify the extent of nuclear polarization relative to ERK signaling and cell morphological change, we measured nuclear polarization as the distance of the nucleus with respect to the cell centroid. The amplitudes of long ERK waves and aspect ratio correlate with the degree of nucleus polarization. Quantile regressions between the 20th and 80th percentile illustrate that heterogeneous ERK and aspect ratio differences show a linear relationship with nuclear polarization (Figure 4D, *p* < 0.0001, 99% confidence intervals shown in Appendix A). To evaluate the temporal changes of the ERK waves and aspect ratios, we compared the average normalized ERK waves and aspect ratios across all fluctuation incidences. Averaging over the bottom 25% of cases where the nuclei are not polarized, ERK signaling barely showed any differences, while the aspect ratio increased slightly and then returned to the original value (Figure 4E). Conversely, in the top 25% of cases with polarized nuclei, ERK signaling reduced over time before increasing after a local minimum at time = 0 (Figure 4E). Correlated with ERK, the aspect ratio increased over time before decreasing after a local maximum at time = 0. We next asked to what extent instantaneous migrating speed relates to fluctuations of long ERK waves and aspect ratio. We measured the instantaneous speed of individual cells and compared it with ERK signaling and aspect ratio when the cell nuclei were polarized. After ERK and aspect ratio passed through a local minimum and maximum, respectively, the cells exhibited a burst of movement when the nuclei are polarized (Figure 4E). On average, this burst of movement had a peak 25% increase in speed ~20 min after the aspect ratio peak. This suggests that nuclear polarization relates to cell migration. Another piece of evidence is that the overall migration speed of individual cells relates linearly to the state of nuclear polarization (Figure 4F). Together, these observations establish that nuclear polarization relates to the degree of ERK reduction, cell elongation, and cell movement burst in speed ~20 min after the local minimum and maximum of ERK and aspect ratios, respectively. This suggests that cells have a cycle of a burst of movement during CXCR4-mediated chemotaxis, in which the extent of ERK reduction, nuclear polarization, cell elongation, and burst in speed are related to each other.

### 3.4. MEK Regulates Both Random Migration and Bursts of Directed Motion, While p38 Regulates Random Migration

To understand to what extent the selected signaling pathways regulate the amplitudes of Akt and ERK, morphology modulations, and resultant cancer cell chemotaxis, we analyzed the dependence of cancer cell responses to targeted inhibitors blocking the oscillatory Akt and ERK network. Alpelisib is a phosphatidylinositol 3-kinase (PI3K) inhibitor, trametinib is a MEK inhibitor, and both are used clinically for cancer therapy. The inhibition of PI3K blocks downstream activation of Akt, while inhibition of MEK blocks the activation of downstream ERK. Kinase p38 has a context-dependent effect on mTORC1 [38], which functions as an inhibitory force in the Akt and ERK oscillatory networks. Inhibiting p38 interrupts the oscillatory networks of Akt and ERK and blocks migration. SB203580 is a specific inhibitor of p38, which is involved in Akt activation and cell survival [39,40,41].

We compared the single-cell data of MDA-MB-231 cancer cells preconditioned with either 100 nM alpelisib, 10 nM trametinib, and 10 µM SB203580 two hours prior to the addition of a CXCL12 gradient (Figure 5A). These concentrations were chosen based on the IC50 values of the compounds (19.3 µM for alpelisib, 1.75 µM for trametinib, and 85.1 µM for SB203580) [41,42,43], their inhibitory effect on overall Akt and ERK levels, and their effect on cancer cell chemotaxis. PI3K inhibition and p38 inhibition, respectively, reduced Akt activity to two times larger to larger than experimental variations (Figure 5B,C). PI3K inhibition, p38 inhibition, and MEK inhibition have progressively increasing inhibitory effects on cancer cell chemotactic responses (Figure 5D). To quantify the effects of the compounds on the random migration and directed motion, respectively, we used a data-driven mean field partial differential equation inference technique called Variational System Identification (VSI) to extract the diffusivity (as a measure of the overall random migration) and advective velocity (as a measure of the directed motion) of cells under each compound inhibition condition. We found that p38 and MEK inhibitors both inhibit random migration of cells, while directed motion is mainly inhibited by trametinib (Figure 5E and Appendix A). Conversely, PI3K inhibition does not affect either the random migration or the directed motion of cancer cells. In relation to the effects of the compounds on random migration and directed motion, we next investigated the effects of inhibitors on the amplitudes of signaling and morphology. This approach identifies how selected signaling pathways regulate signaling and morphology amplitudes and informs the development of therapies targeting cancer cell chemotaxis.

We compared the effects of these compounds on various aspects of signaling and morphology amplitudes in relation to random migration and directed motion of cancer cells. We focused on amplitudes of short ERK and Akt waves relative to the heterogeneity of random migration. Despite similar reductions in mean Akt levels and no reduction in mean ERK levels (Figure 5B,C), the amplitudes of short signaling waves have different reductions in response to PI3K inhibition and p38 inhibition. The short ERK and Akt wave amplitudes are negligibly to minimally reduced under PI3K inhibition but minimally to modestly reduced under p38 inhibition (Figure 5F,G). This is consistent with the negligible and modest reduction in random migration under PI3K and p38 inhibition, respectively (Figure 5E). As for MEK inhibition, the amplitudes of short ERK and Akt waves, consistent with random migration of the cells, have modest to strong reductions (Figure 5F,G). Conversely, we focused on the amplitudes of long ERK/Akt waves and aspect ratio, and nuclear polarization relative to the heterogeneity of directed motion. PI3K and p38 inhibitions have negligible and minimal reductions in these amplitudes or nuclear polarization (Figure 5F,G), which is consistent with the similar minimal change in directed motion with these inhibitors. Conversely, consistent with the inhibition of directed motion, trametinib produced modest to strong reductions in amplitudes of long Akt/ERK waves and nuclear polarization (Figure 5F,G). Furthermore, we evaluated the effects on the periodic burst of speed in directed motion. The cells under PI3K inhibition still have a similar periodic burst of migration speed ~20 min after morphological elongation (Figure 5H). Consistent with reduced overall migration speed under p38 inhibition, p38 inhibition resulted in a smaller burst of migration speed. However, the periodic burst of speed is impaired by MEK inhibition, as evidenced by a lack of speed change. The ERK signaling local minima for cells under MEK inhibition are also less aligned with the aspect ratio maxima (Appendix A). Together, these observations led us to two conclusions: First, the MEK-ERK signaling pathway regulates both overall motive force and directed movement due to the CXCL12 gradient, while p38 signaling regulates only the overall motive force of the cell. PI3K-Akt signaling pathway does not seem to regulate either the overall motive force or the directed movement of cells. Second, the MEK-ERK signaling pathway regulates nuclear polarization and periodic bursts of cell movement.

## 4. Discussion

The present study investigated to what extent oscillatory signaling networks in cells are associated with heterogeneous chemotactic responses and the effects of targeted inhibitors. This single-cell chemotactic study in MDA-MB-231 cells led us to two main conclusions. First, the heterogeneity of cancer cell directed motion and random migration originates from the rate of change of kinase activities and cell morphology with oscillation period as the regulator. Second, the responsiveness of oscillatory signals and chemotaxis is regulated by the activities of MEK and p38 kinases. The fact that the heterogeneity of chemotactic response in cancer cells is determined by the oscillatory network and can be controlled by inhibition of signaling pathways suggests that potential treatment approaches for targeting fast-migrating cells should consider effects on oscillatory signaling and morphology.

Three analyses provide evidence that the heterogeneity of cancer cell directed motion and random migration originate from the rate of change of signaling activities and cell morphology with oscillation period as the regulator. First, the migration speeds of individual cells are determined by the amplitude of Akt and ERK signaling waves and the morphological aspect ratio of each cell. Despite not referring to cell-to-cell heterogeneity, other previous investigations reinforce that ERK pulses are related to oncogenic transformation, collective migration, and polarization [20,21,22]. On the other hand, previous observation also demonstrates that cells with large lateral elongations have higher migration speeds [44]. Together, these studies lend further credence to our earlier suggestion that the responsiveness of individual cells depends on pre-existing states that regulate the rate of change rather than the absolute level of signaling. In our previous studies, we identified pre-existing states of individual cells that regulate heterogeneous Akt and ERK signaling responses to CXCR4 and EGFR signaling [9,10]. Other groups have also identified pre-existing cellular states that affect the signaling decisions in single cells [14,45]. For instance, cellular states change when cells undergo epithelial-mesenchymal transition and acquire higher migratory potential [46]. Other investigators and we have focused on the change of signaling level after adding the bolus of ligand stimulation as the responsiveness of the cell rather than the absolute signaling level [9,10,45,46,47]. In the context of this current study, the change in the signaling level is the same as the amplitude of signaling with an “ultra-long” oscillation period. When the investigation shifts towards the migratory and morphological responses of individual cells, this study shows that we should focus on the rate of change of signaling and morphological change with shorter oscillation periods. This study also supports previous studies that cells encode signaling networks using two main mechanisms. Amplitude modulations refer to the idea that the actual concentration of signaling molecules encodes the stimulus, whereas frequency modulations describe the mechanism where the period between consecutive bursts represents the stimulus [48]. However, ERK pulses cannot be easily detected in cells with mutations in the Ras-ERK pathway, such as the MDA-MB-231 cells. The dynamic signal transmission properties are altered such that short pulses of input Ras activity are distorted into abnormally long ERK outputs [29]. To resolve the rate of change of signaling in MDA-MB-231 cells with K-Ras mutation, this study uses bandpass filtering to quantify the signaling fluctuations in the form of signaling amplitudes within certain oscillation period ranges. This method has the advantage of independently separating amplitude and frequency modulations. Our study suggests that these signaling and morphological amplitudes are alternative ways that MDA-MB-231 cells with a K-Ras mutation encode information related to migratory response.

The second piece of evidence supports that the oscillation period is a regulator of the signaling and morphological amplitudes for directed motion and random migration heterogeneity. Depending on the oscillation periods of Akt and ERK signaling waves and morphological aspect ratio, cells have distinct levels of dependence on the CXCL12 gradient. Our current study shows that short Akt and ERK signaling waves with an oscillation period of 0.5–1 h contribute to random migration, whereas long signaling waves and morphological change with an oscillation period of 1.5–2 h tend to contribute to directed motion. This suggests that cell signaling networks regulate distinct modes of migration via different time scales, which was suggested previously [49]. Random migration relates to random actin protrusions in a cell, which were found previously to relate to ERK pulses with similar oscillation periods of the short Akt and ERK signaling waves in this study [19,20,21,22]. Conversely, directed motion relates to the polarization and reorientation of the nucleus, which happen on a longer time scale [37,50]. This evidence supports that the rate of change of signaling activities and cell morphology separately determine directed motion and random migration of individual cells when the oscillation period is long versus short, respectively.

The third piece of evidence further supports that the rate of change of long ERK signaling waves and morphological aspect ratio with similar 1.5–2 h oscillation periods negatively correlate to determine the directed motion of cancer cells. Our current study shows a relationship between the extent of reduced long ERK signals and the extent of cell elongation. Cells reduce the long ERK signal and elongate at the same time and have a burst of migration speed with a delay of around 20 min. This evidence and previous investigations on short ERK pulses [19,20] demonstrate that ERK waves connect to feedback networks involving distinct aspects of cytoskeletal remodeling depending on ERK oscillation periods. Other previous studies also show that ERK initiates and reinforces cellular protrusions so that cells have sustained forward motion at the leading edge [51,52]. This observation explains that cancer cells with periodic ERK activity have periodic cell morphology and migration speed. Our study also lends credence to previous studies that spontaneous waves of ERK activation in collective migration alter cell mechanics by modulating actomyosin contractility [21]. The change in cell mechanics by long ERK waves could be one of the reasons why cells have bursts of speed during directed motion.

Three analyses provide evidence that activities of MEK and p38 kinases regulate the responsiveness of oscillatory signals and chemotaxis. First, the state of MEK kinase activity determines the responsiveness of both short and long signaling and morphological amplitudes with resultant impairment of cancer cell random migration and directed movement. Our study shows that the amplitudes of short and long ERK and Akt waves and aspect ratios reduce when cells are treated with a MEK inhibitor. A previous study showed that inhibitory phosphorylation of MEK is crucial for negative feedback and oscillatory ERK signaling and impacts signaling amplitudes [23]. Furthermore, another study on cell mechanics demonstrated that MEK inhibition suppresses focal adhesions and cell spreading [53]. These findings support our results regarding the effects of MEK on the amplitudes of signaling and morphological oscillations. The effects of targeted inhibitors on random migration and directed motion cannot be easily identified because cells have both directed motion and random migration during chemotaxis. To delineate these mechanisms of cell motion, we used VSI to quantitatively infer the directed motion and random migration from the experimental data. This approach rests on the result, classically established in statistical physics, that the random migration of individual cells leads to the mechanism of diffusion in a cell population. Analogously, directed motion leads to advection. While diffusion causes the multi-directional spreading of the density of cells, advection carries the density along the vector of the directed motion. The time-dependent, advection–diffusion partial differential equation separates these mechanisms into distinct differential terms while representing their combined effect as the time derivative of the cell density. Using the spatio-temporal data on cell dynamics under control and drug conditions, this partial differential equation can be inverted to infer the presence and relative strength of diffusion and advection and, therefore, of random migration and directed motion in chemotaxis. The results of our data-driven inference show that MEK inhibition reduces both directed motion and overall random migration of the cells. It is notable that the two completely independent analytical methods in this study—signaling/morphology amplitudes and VSI—concluded the same effect of MEK on cell migration. While in other biological contexts, previous studies also support that MEK blocks both directed motion [54] and overall migration [55]. Together, this study and previous investigations support the idea that MEK kinase activity impacts not onky absolute ERK activity but also the amplitudes of signaling and morphology oscillations, thereby regulating chemotaxis.

Second, MEK inhibition also impacted nuclear polarization, which relates to the extent of ERK signal change, cell elongation, and directed motion acceleration. This study shows that nuclear polarization relates to single-cell changes in ERK signaling, cell elongation, and directed motion. This evidence supplements previous investigations that nuclear polarization promotes polarity and directed motion in cells [37,50]. However, to our knowledge, there were no prior reports of MEK regulating nuclear polarization. With our multimodal single-cell analysis, we are limited to identifying the causal relationships among nuclear polarization, ERK signaling change, cell elongation, and directed motion. Therefore, we conclude that nuclear polarization, which promotes cell elongation and directed motion, is related to ERK signaling and is affected by MEK inhibition.

Third, the state of p38 kinase activity determines the responsiveness of only short Akt and ERK signaling amplitudes and blocks the random migration of cancer cells. Our study shows that the amplitudes of short ERK and Akt waves reduce when cells are treated with a p38 inhibitor. Recognizing that p38 may have context-dependent effects on mTorc1, our past work with these cells shows that p38 inhibits mTORC1 [38,56,57]. Since mTORC1 functions as a central negative regulator of signaling by Akt and ERK, blocking p38 increases the inhibitory effect of mTORC1 on the Akt and ERK pathways. This relationship provides a mechanism for decreases in Akt and ERK signaling amplitudes when cancer cells were preconditioned with p38 inhibitor prior to chemotaxis. This result is consistent with past work showing that p38 promotes cell random migration [58], as evidenced by the effects of p38 on actin polarization and dynamics [59,60]. Together, this study and previous investigations establish that p38 kinase activity impacts the amplitudes of short signaling oscillations and actin dynamics, which determine the random migration of cells.

This study assumes that live cell imaging reporters in cells capture all essential temporal information. However, the characterization of network dynamics is limited by imaging frequency and types of live-cell reporters. Our assumption is reasonable because the imaging frequency is five times faster than the signal dynamics that we studied. With recent advances in automated imaging techniques and live-cell reporters, the imaging frequency and the number of live-cell reporters can be increased in the future without causing phototoxicity or fluorescence crosstalk. This improvement will help reveal other mechanisms regulating heterogeneity in cancer cells, such as receptor dynamics, cytoskeletal structure, adhesion, and contractility. Cells also migrate differently in 2D versus 3D environments. We will need to confirm to what extent discoveries in this 2D chemotaxis study apply to a 3D environment. Our quantitative analysis approach focused on relationships and correlations among signaling, morphology, and migration and variations between migrating and non-migrating cells. Further studies are required to firmly establish causal relationships between these parameters. Future studies also are needed to explore pre-existing cell states and unique feedback delays in Ras/PI3K/ERK signaling and morphological networks regulating migration heterogeneity. Our study offers answers to processes in oscillatory signaling that associate with the heterogeneous responses of seemingly identical cancer cells with the same genetic mutation. In addition to the heterogeneity described here, heterogeneity also arises within a tumor and across different patients. While our study provides insights about the effective targeting of migrating cancer cells with constitutive activation of one major pathway, our findings need to be combined with other studies focusing on intratumor heterogeneity for effective prevention of metastasis.

In summary, we have shown that the rate of change of ERK and Akt signaling activities and cell morphology determine the heterogeneity of cancer cell chemotaxis. We also demonstrated the effects of MEK and p38 kinases on random and directional cell movement. Our study suggests that future studies in signaling, morphology, and chemotaxis should consider the period and amplitude of oscillatory signaling outputs. Our finding paves the path towards better understanding of mechanisms regulating heterogeneous responsiveness in cancer cells and future identification of new interventions controlling cancer metastasis.

## Figures and Tables

**Figure 1 bioengineering-10-00269-f001:**
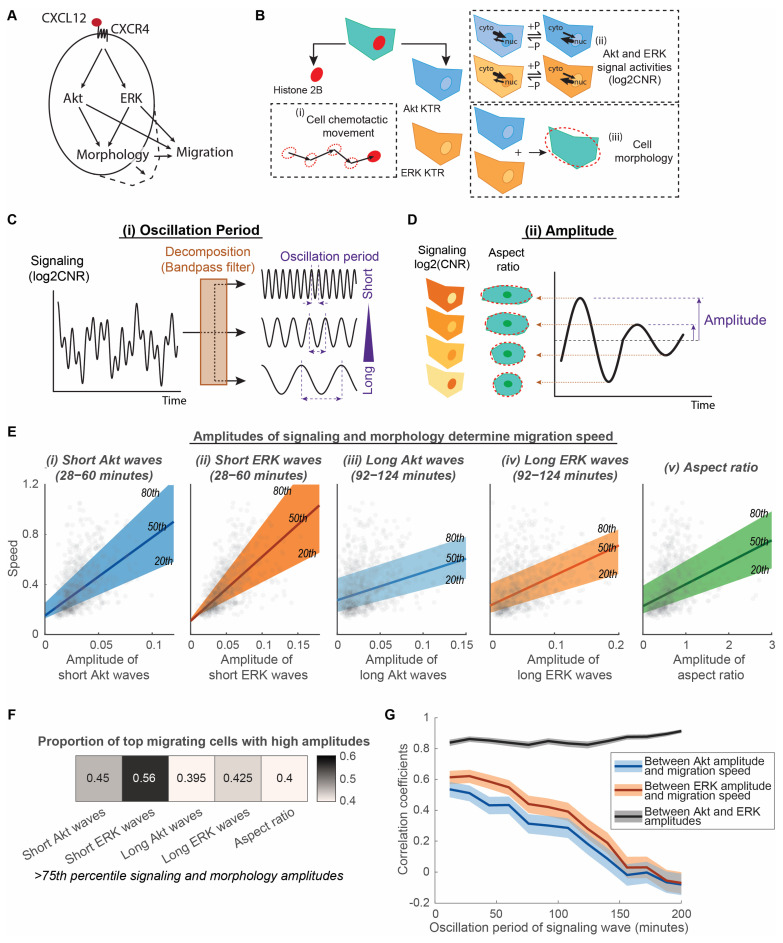
Amplitudes of Akt and ERK signaling waves and morphological aspect ratio determine migration speed of cancer cells. (**A**). Processes connecting signaling and migration. CXCL12 binds to CXCR4 and elicits downstream Akt and ERK signaling, changes in cell morphology, and cell migration. (**B**). Quantification of heterogeneous signaling, morphology, and migration in individual cells. Cells express three fluorescence reporters: histone 2B marker, Akt kinase translocation reporter (KTR), and ERK KTR. These fluorescence reporters were used to quantify (i) cell chemotactic movements, (ii) Akt and ERK signaling activities, and (iii) cell morphology. Phosphorylation (+P) and dephosphorylation (−P) of the kinase substrate drives the reporter into the cytoplasm (kinase “on”) or nucleus (kinase “off”), respectively. The quantification of kinase activity uses log2 cytoplasmic-to-nuclear ratios (CNR) of the reporter fluorescence intensity. (**C**). Decomposition of time-lapse data into simple sinusoidal waves with different periods. The time-lapse data include Akt and ERK signaling levels (with a unit of log2CNR). The time-lapse data are decomposed using bandpass filtering into multiple signaling waves with distinct oscillation periods. (**D**). Measurement of signaling and morphological amplitudes. The magnitude of the time-lapse oscillation data is quantified as amplitude. The time-lapse oscillation data includes Akt and ERK signaling waves after decomposition from C and morphological aspect ratio. The average amplitude of the oscillation data was calculated for each individual cell. (**E**). Linear relationship of speed with amplitudes of signaling and morphology. The x-axes of the five plots illustrate the amplitudes of (i) short Akt waves, (ii) short ERK waves, (iii) long Akt waves, (iv) long ERK waves, and (v) aspect ratio. The y-axes are all migration speeds. The 20th, 50th, and 80th quantile regression results were overlayed to illustrate the relationship among signaling, morphology, amplitude, and speed. n_c_ = 799. (**F**). Proportion of top migrating cells with high amplitudes. Heatmap showing the proportion of fast migrating cells (>75th percentile) that have high signaling and morphology amplitudes (>75th percentile). nc = 799. (**G**). Correlation coefficients between Akt amplitude, ERK amplitude, and migration speed are plotted with varying oscillation periods for the signaling waves. The shaded area is the 95% confidence interval for correlation coefficients. Higher correlation coefficient indicates stronger linear relationship between the two determinant factors. n_c_ = 799.

**Figure 2 bioengineering-10-00269-f002:**
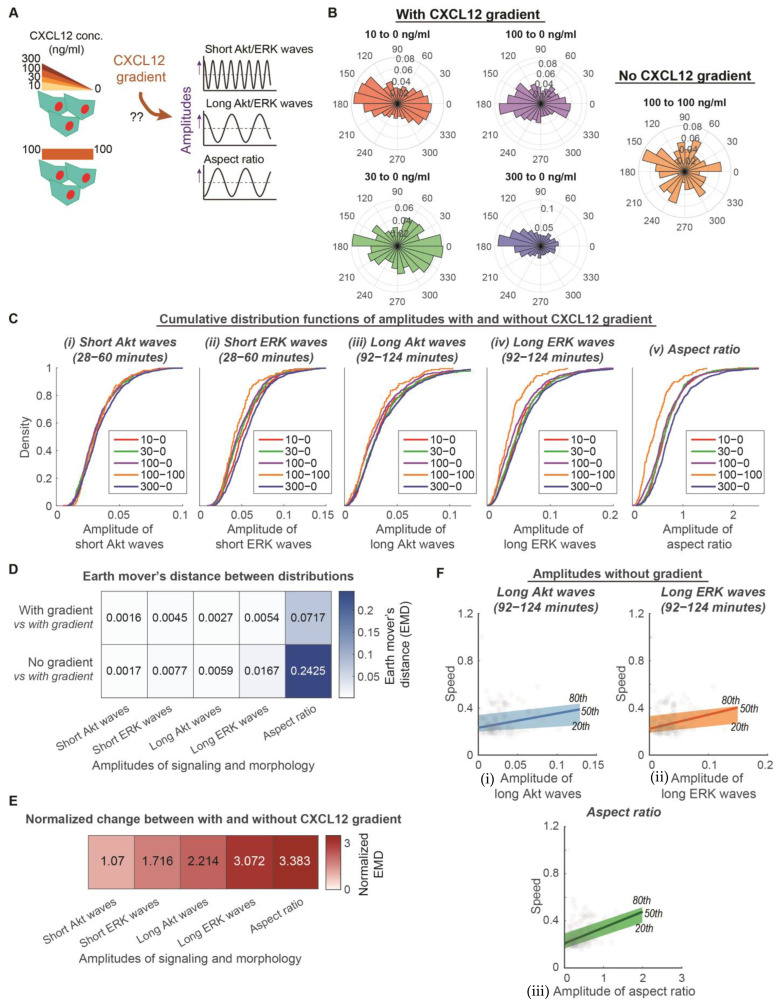
Amplitudes of long Akt and ERK waves and aspect ratio determine heterogeneity of directed movement, whereas amplitudes of short Akt and ERK waves determine heterogeneity of overall motive force. (**A**). Experimental design to determine the extent that CXCL12 gradients regulate amplitudes of signaling and morphology. MDA-MB-231 cells are exposed to CXCL12 stimulations with various levels of gradient (10 ng/mL to 0 ng/mL, 30 ng/mL to 0 ng/mL, 100 ng/mL to 0 ng/mL, and 300 ng/mL to 0 ng/mL) and no gradient (100 ng/mL to 100 ng/mL). (**B**). Migration behavior of cells in a variety of CXCL12 gradient and no gradient environments. Polar histograms, which describe the overall direction of cancer cell movement, are plotted. Red: 10 to 0 ng/mL of CXCL12 (n_c_ = 581); Green: 30 to 0 ng/mL of CXCL12 (n_c_ = 495); Purple: 100 to 0 ng/mL of CXCL12 (n_c_ = 799); Blue: 300 to 0 ng/mL of CXCL12 (n_c_ = 1224); Orange: 100 to 100 ng/mL of CXCL12 (n_c_ = 158). (**C**). Cumulative distribution functions of various aspects of signaling and morphology amplitudes among conditions with various levels of gradient and without gradient. X-axis of each plot is the aspects of signaling and morphology amplitudes: Amplitude of short Akt waves, amplitude of short ERK waves, amplitude of long Akt waves, amplitude of long ERK waves, and amplitude of aspect ratio. Legend in each plot indicates the different conditions with various levels of gradient (10 ng/mL–0 ng/mL (n_c_ = 581), 30 ng/mL–0 ng/mL (n_c_ = 495), 100 ng/mL–0 ng/mL (n_c_ = 799), 300 ng/mL–0 ng/mL (n_c_ = 1224)) and without gradient (100 ng/mL–100 ng/mL (n_c_ = 158)). (**D**). Average earth mover’s distance (EMD) between two distributions in which both have gradient versus one of which does not have a gradient. (**E**). Effect of gradient on signaling and morphology amplitudes. The normalized EMD from D was calculated to show the effect of gradient. Darker color indicates that the aspect of signaling or morphology amplitude has a stronger dependence on gradient. (**F**). Linear relationship of speed with amplitudes of long signaling waves and morphology in conditions without a CXCL12 gradient (n_c_ = 158). The x-axes of the three plots illustrate the amplitudes of (i) long Akt waves, (ii) long ERK waves, and (iii) aspect ratio. Their y-axes are all migration speed. The 20th, 50th, and 80th quantile regression results were overlayed to illustrate the poor relationship between signaling/morphology amplitude and speed without CXCL12 gradient.

**Figure 3 bioengineering-10-00269-f003:**
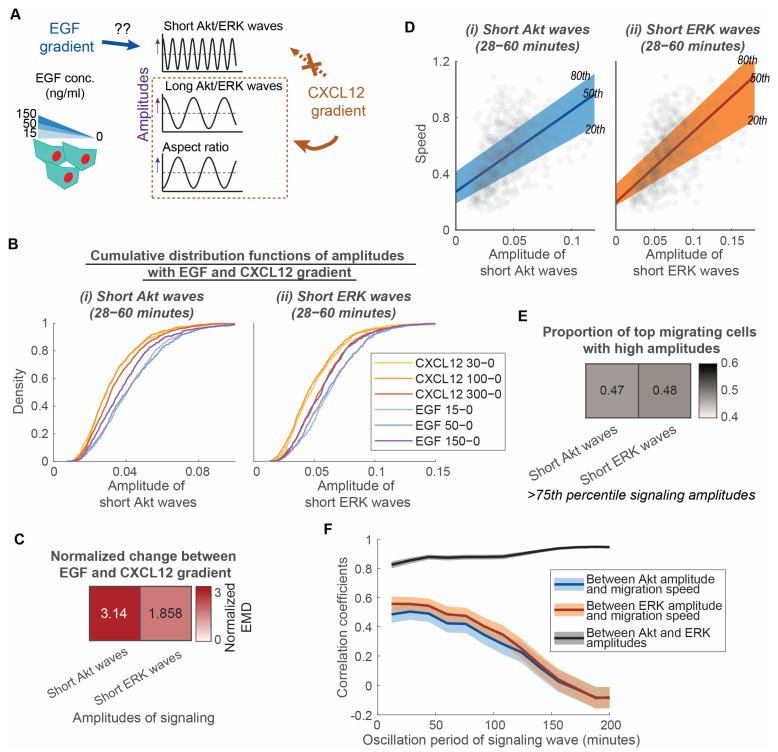
Short Akt waves and short ERK waves have distinct roles in determining the heterogeneity of overall motive force in different classes of receptor-mediated chemotaxis. (**A**). Experimental design to determine the extent that EGF gradients regulate the amplitude of short Akt and ERK waves compared with a CXCL12 gradient. MDA-MB-231 cells are exposed to EGF stimulations with various levels of gradient (15 ng/mL to 0 ng/mL, 50 ng/mL to 0 ng/mL, 150 ng/mL to 0 ng/mL). (**B**). Cumulative distribution functions of various aspects of signaling and morphology amplitudes among conditions with various levels of CXCL12 gradients and EGF gradients. X-axis of each plot is the aspects of signaling and morphology amplitudes: Amplitude of short Akt waves and amplitude of short ERK waves. Legend in each plot indicates the different conditions with CXCL12 gradients (CXCL12 30 ng/mL–0 ng/mL (n_c_ = 495), CXCL12 100 ng/mL–0 ng/mL (n_c_ = 799), CXCL12 300 ng/mL–0 ng/mL (n_c_ = 1224)) and with EGF gradients (EGF 15 ng/mL–0 ng/mL (n_c_ = 721), EGF 50 ng/mL–0 ng/mL (n_c_ = 548), EGF 150 ng/mL–0 ng/mL (n_c_ = 560)). (**C**). Normalized earth mover’s distance (EMD) between distributions with CXCL12 gradients and EGF gradients. Darker color indicates that the aspect of signaling amplitude has a stronger dependence on the EGF gradient versus CXCL12 gradient. (**D**). Linear relationship of speed with amplitudes of short Akt and ERK waves in conditions with 15 ng/mL to 0 ng/mL EGF gradient (n_c_ = 721). The x-axes of the two plots illustrate the amplitudes of (i) short Akt waves and (ii) short ERK waves. Their y-axes are all migration speed. The 20th, 50th, and 80th quantile regression results were overlayed to illustrate the relationship between signaling amplitude and speed. (**E**). Proportion of top migrating cells with high amplitudes in conditions with 15 ng/mL to 0 ng/mL EGF gradient (n_c_ = 721). Heatmap showing the proportion of fast migrating cells (>75th percentile) that have high short Akt and ERK wave amplitudes (>75th percentile). (**F**). Correlation coefficients between Akt amplitude, ERK amplitude, and migration speed are plotted with varying oscillation periods for the signaling waves in conditions with 15 ng/mL to 0 ng/mL EGF gradient (n_c_ = 721). The shaded area is the 95% confidence interval for correlation coefficients. The overlap of shaded area indicates insignificance difference in the correlation coefficient.

**Figure 4 bioengineering-10-00269-f004:**
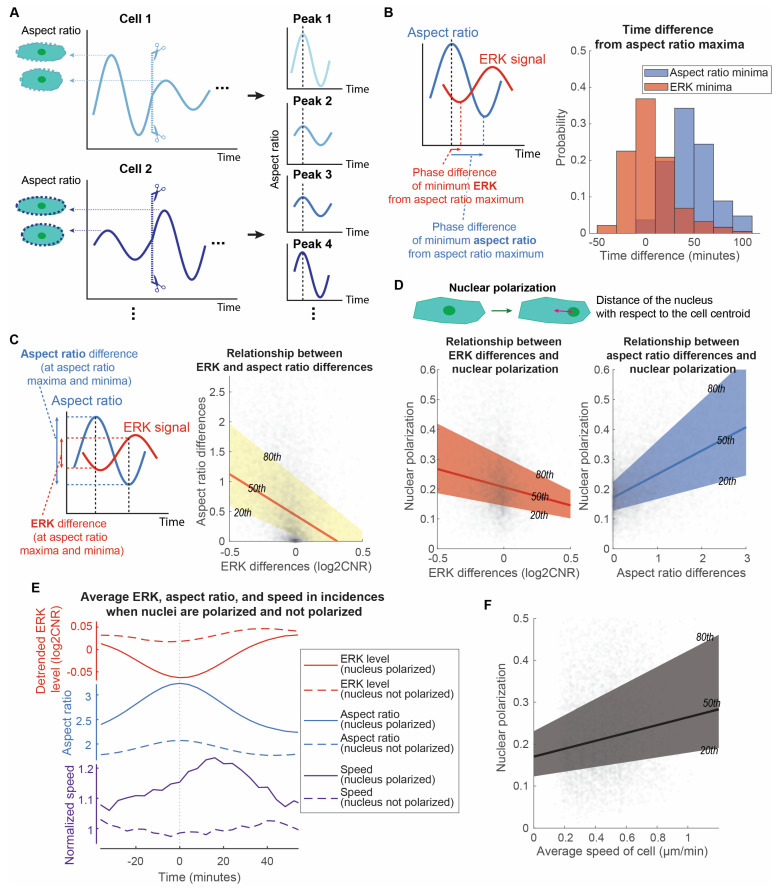
Cell directed motion relates to the extent of long ERK wave reduction, cell elongation, and nuclear polarization. (**A**). Analytic method to isolate and align each aspect ratio peak of each cell. (**B**). Distributions of aspect ratios and ERK time differences. With the aspect ratio peak aligned, the time differences from this peak to the local minima of aspect ratio and ERK signaling were measured. The overall ERK signal was detrended to eliminate the overall ERK trend and focus on the oscillation patterns. The distributions of the time differences for aspect ratio and ERK minima were illustrated in a histogram (n_p_ = 5509). (**C**). Linear relationship of ERK differences and aspect ratio differences. At the local maxima and minima of aspect ratio, the aspect ratio and ERK levels were measured to compute the difference between the local maxima and minima. The x-axis and y-axis of the right plot are the ERK differences and aspect ratio differences, respectively (n_p_ = 5498). The 20th, 50th, and 80th quantile regression results were overlayed to illustrate the relationship between ERK and aspect ratio differences. (**D**). Linear relationship of nuclear polarization with ERK difference and aspect ratio difference, respectively. The x-axes of the two plots illustrate the ERK difference and aspect ratio difference at the local maxima and minima of aspect ratio. Their y-axes are all nuclear polarization (n_p_ = 5498). We defined the nuclear polarization using the distance of the nucleus with respect to the cell centroid normalized to the minor axis length of the cell. The 20th, 50th, and 80th quantile regression results were overlayed to illustrate the relationship between nuclear polarization and ERK/aspect ratio differences. (**E**). Mean temporal changes of ERK, aspect ratio, and speed of cells in incidences when their nuclei are polarized versus not polarized. The plot is divided into three parts to illustrate the temporal changes: detrended ERK level (red), aspect ratio (blue), and normalized speed (purple). Solid lines represent incidences when nuclei are polarized (n_p_ = 1375), while dotted lines represent incidences when nuclei are not polarized (n_p_ = 1374). Instantaneous speed of cells is normalized with respect to the overall speed of the cells and average speed when nuclei are not polarized. (**F**). Linear relationship of average speed of cell and nuclear polarization. The x-axis and the y-axis are the average speed of cells throughout the length that we have tracked them and the nuclear polarization during elongation of cells, respectively (n_p_ = 5498).

**Figure 5 bioengineering-10-00269-f005:**
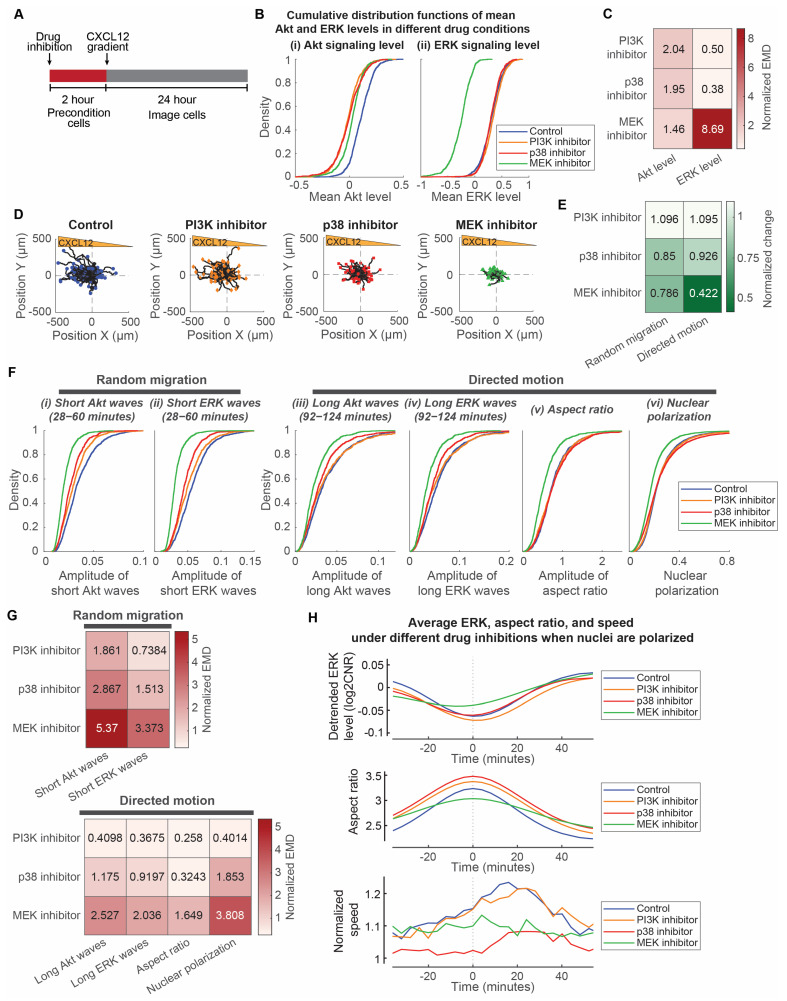
MEK regulates both random migration and bursts of directed motion, while p38 regulates random migration. (**A**). Experimental design to precondition cells with inhibitors and image chemotaxis toward a CXCL12 gradient. (**B**). Cumulative distribution functions of mean Akt and ERK levels treated with different compounds. X-axes of the two plots represent the mean Akt level and mean ERK level, respectively. Legend in each plot indicates the different inhibitor conditions: Control (no compound; n_c_ = 1224), PI3K inhibitor (100 nM alpelisib; n_c_ = 742), p38 inhibitor (10 µM SB203580; n_c_ = 785), and MEK inhibitor (10 nM trametinib; n_c_ = 889). (**C**). Normalized earth mover’s distance (EMD) between the mean Akt and ERK level distributions of inhibitor conditions with respect to the control. Darker color indicates that the mean signaling levels have a larger difference from the control. (**D**). Migratory trajectories of cells treated with different inhibitors. 100 randomly selected single-cell trajectories (black lines) are normalized to the origin and are shown with different colors/shape for the end position of each cell under various conditions. Blue circle: control (n_c_ = 1224); orange diamond: PI3K inhibitor 100 nM alpelisib (n_c_ = 742); Red rectangle: p38 inhibitor 10 µM SB203580 (n_c_ = 785); Green triangle: MEK inhibitor 10 nM trametinib (n_c_ = 889). (**E**). Normalized change of random migration and directed motion in response to targeted inhibitors. Variational system identification was used to quantify the random migration and directed motion of the overall cell population. Random migration was normalized from the diffusivity of cell. Directed motion was normalized from the ratio of migrated displacement along the direction of gradient to the overall migrated displacement. (**F**). Cumulative distribution functions of aspects of signaling and morphology that relate to random migration and directed motion with different inhibitors. Aspects of signaling and morphology include short Akt and ERK wave amplitudes, long Akt and ERK wave amplitudes, aspect ratio amplitudes, and nuclear polarization. Legend in each plot indicates the different inhibitor conditions: Control (no compound; n_c_ = 1224), PI3K inhibitor (100 nM alpelisib; n_c_ = 742), p38 inhibitor (10 µM SB203580; n_c_ = 785), and MEK inhibitor (10 nM trametinib; n_c_ = 889). (**G**). Normalized earth mover’s distance (EMD) between aspects of signaling and morphology distributions of inhibitor conditions with respect to the control. Aspects of signaling and morphology include short Akt and ERK wave amplitudes, long Akt and ERK wave amplitudes, aspect ratio amplitudes, and nuclear polarization. Darker color indicates a larger difference from the control. (**H**). Differences between mean temporal changes of ERK, aspect ratio, and speed of cells in different inhibitor conditions when cell nuclei are polarized. The plot is divided into three parts to illustrate the temporal changes: detrended ERK level (**top**), aspect ratio (**middle**), and normalized speed (**bottom**). The instantaneous speeds of cells are normalized with respect to the overall speed of the cells and average speed when nuclei are not polarized. Legend in each plot indicates the different drug inhibition conditions when nuclei are polarized: Control (no compound; n_p_ = 1375), PI3K inhibitor (100 nM alpelisib; n_p_ = 758), p38 inhibitor (10 µM SB203580; n_p_ = 932), and MEK inhibitor (10 nM trametinib; n_p_ = 898).

## Data Availability

The data presented in this study are available on request from the corresponding author.

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
