# Peer review of "Oscillatory ERK Signaling and Morphology Determine Heterogeneity of Breast Cancer Cell Chemotaxis via MEK-ERK and p38-MAPK Signaling Pathways"

_bioengineering, 2023, doi:10.3390/bioengineering10020269_

Round 1

Reviewer 1 Report

The authors uncovered the role of ERK in breast cancer.

The manuscript is of interest.

Points to be addressed:

1) The rationale of why the authors came up with this research is scanty and is related to a lack of novelty: please highlight what this manuscript might add.

2) What is the information that is not exactly available that motivated the authors to come up with this information. What are the current caveats and how do the authors highlight the current research in answering them? If not they need to address in background and infuture directions .

3)State of the art figures are required: scale bar should be provided in high resolution.

4)The authors could provide a little more consideration of genomic directed stratifications in clinical trial design and enrolments. 

5)The underlying message here is that more precision and individualized approaches need to be tested in well-designed clinical trials – a challenge, but I would be interested in their perspective of how this might be done. If beyond the scope of the manuscript, this should be highlighted as a limitation

6)Level of consistency among studies should be better reported; please provide

7)The authors need to highlight what new information the review is providing to enhance the research in progress; the authors themselves mentioned and quote (ref.21) the role of CXCL12: in introduction  and discussion sections this reviewer personally misses few insights regarding the role of microenvironment in mediating key biological steps in Breast cancer metastatization in crucial site (i.e. bone):  different approaches have been proposed in order to resolve the biologic complexity underlying the pathophysiologic step to bone dissemination. The first of the steps in the process of cancer spreading to the bone is the homing to the marrow microenvironment throughout the bloodstream of the tumor cells, via the neo-angiogenesis process, trough permissive bone marrow endothelial cells. Remarkably, a prone microenvironment is involved in the cancer cycle, educating and hijacking the tumor niche that is to be colonized throughout a neoplastic permissive environment. Ancillary to the cancer intrinsic mechanisms, peri-neoplastic infiltrates actively prime drug-resistance mechanisms both in solid and hematologic neoplasms expressing an osteotropic phenotype. Therefore, a number of molecular actors have been considered as elements that drive the neoplastic cells to the bone environment, including karyotypic non-random abnormalities (i.e., t(11:22), t(15;19), t12p, t(X;18), and del11p), plasma membrane protrusions, cytoskeleton systems, adhesion molecule systems such as CXCL12/CXCR4 (chemokine ligand 12/chemokine receptor 4), junctional adhesion molecules in osteotropic tumors , focal adhesion kinases, and vascular and immune-microenvironment interactions

 (please refer to PMID: 31470608 and expand). This can boost the discussion and envision potential novel therapeutic windows, especially in particularly aggressive phenotype of breast cancer and other correlated neoplasms (i.e. SMUP).

Author Response

Summary: The authors uncovered the role of ERK in breast cancer. The manuscript is of interest.

Point 1: The rationale of why the authors came up with this research is scanty and is related to a lack of novelty: please highlight what this manuscript might add.

Response 1: Thank you Reviewer 1 for the suggestions. CXCL12-CXCR4 and EGF-EGFR are important signaling molecules for metastasis in breast cancer. However, targeting them has been difficult since there is marked heterogeneity in activation of intracellular signaling and chemotaxis among seemingly identical cancer cells. Chemotaxis plays an important role in the migration of cancer cells and subsequent metastasis towards chemotactic molecules CXCL12 and EGF. Chemotaxis is regulated by signaling oscillations in oncogenic ERK and Akt pathways. However, due to incomplete understanding of processes underlying oscillatory signaling networks that regulate heterogeneous chemotactic responses of cancer cells, therapeutic efforts to block dissemination of cancer cells and subsequent metastasis remain stalled. In this study, we employed an integrated approach combining live, single-cell imaging and quantitative analysis to answer to what extent oscillatory signaling networks in cells associate with heterogeneous chemotactic response and effects of targeted inhibitors.

Point 2: What is the information that is not exactly available that motivated the authors to come up with this information. What are the current caveats and how do the authors highlight the current research in answering them? If not they need to address in background and infuture directions .

Response 2: Adding to the above response, current research lacks a focus on understanding processes in oscillatory signaling networks that associate with heterogeneous chemotactic responses of cancer cells. What processes lead to some cancer cells to migrate towards a chemotactic gradient versus some that do not? Do cancer cells have different processes connecting oscillatory signaling, morphology, and chemotaxis across different timescales? These questions remain poorly defined as we have described in the first three paragraphs of the revised manuscript. Our live, single-cell imaging and quantitative analysis approach employing time-dependent data processing and variational system identification provides more detailed answers on how cells decode these pathways and chemotax. 

Point 3: State of the art figures are required: scale bar should be provided in high resolution.

Response 3: Graphical abstract is attached with the revised manuscript. We will make sure the resolution of our figures meets the standards of the journal.

Point 4: The authors could provide a little more consideration of genomic directed stratifications in clinical trial design and enrolments.

Response 4: Clinical trials in breast cancer and other malignancies are moving towards personalized medicine, matching a new targeted compound to patients with mutations in the corresponding drug target and pathway. For example, depending on specific mutations present in a tumor, ongoing clinical trials in triple negative breast cancer are testing compounds against PI3K pathways and/or MAPK pathways. Downstream Akt and ERK kinases are important determinants of the activation state of these pathways. We have live-cell kinase reporters monitoring Akt and ERK. Heterogeneity is common across different patients as well as within a tumor. Current practice uses biopsy and circulating DNA to determine therapy targeting the dominant mutation. Often cancer cells with different mutations escape the treatment and cause progressive or recurrent disease. We and others found that heterogeneity also arises in seemingly identical cancer cells due to distinct preexisting cell states. Our work provides a better understanding of processes regulating heterogeneous responses in cancer cells and potential future interventions controlling cancer metastasis.

Point 5: The underlying message here is that more precision and individualized approaches need to be tested in well-designed clinical trials – a challenge, but I would be interested in their perspective of how this might be done. If beyond the scope of the manuscript, this should be highlighted as a limitation

Response 5: Adding to response 4, our study provides detailed information about processes associated with heterogeneity of cancer cells with constitutive activation of one major pathways. Our findings need to be combined with other studies that focus on intratumor heterogeneity for effective design of clinical trials and prevention of metastasis. We have added information regarding this in the second last paragraph of the discussion section.

Point 6: Level of consistency among studies should be better reported; please provide

Response 6: In this study, we focus on the heterogeneous response of cancer cells. Therefore, we are mainly comparing behaviors of single cells. In our study, different studies with the same conditions are combined to analyze the behavior of individual cells. In the revised manuscript, we report the number of cells or the number of aspect ratio peaks in each figure legend. We described this method in the Methods section.

Point 7: The authors need to highlight what new information the review is providing to enhance the research in progress; the authors themselves mentioned and quote (ref.21) the role of CXCL12: in introduction  and discussion sections this reviewer personally misses few insights regarding the role of microenvironment in mediating key biological steps in Breast cancer metastatization in crucial site (i.e. bone):  different approaches have been proposed in order to resolve the biologic complexity underlying the pathophysiologic step to bone dissemination. The first of the steps in the process of cancer spreading to the bone is the homing to the marrow microenvironment throughout the bloodstream of the tumor cells, via the neo-angiogenesis process, trough permissive bone marrow endothelial cells. Remarkably, a prone microenvironment is involved in the cancer cycle, educating and hijacking the tumor niche that is to be colonized throughout a neoplastic permissive environment. Ancillary to the cancer intrinsic mechanisms, peri-neoplastic infiltrates actively prime drug-resistance mechanisms both in solid and hematologic neoplasms expressing an osteotropic phenotype. Therefore, a number of molecular actors have been considered as elements that drive the neoplastic cells to the bone environment, including karyotypic non-random abnormalities (i.e., t(11:22), t(15;19), t12p, t(X;18), and del11p), plasma membrane protrusions, cytoskeleton systems, adhesion molecule systems such as CXCL12/CXCR4 (chemokine ligand 12/chemokine receptor 4), junctional adhesion molecules in osteotropic tumors , focal adhesion kinases, and vascular and immune-microenvironment interactions (please refer to PMID: 31470608 and expand). This can boost the discussion and envision potential novel therapeutic windows, especially in particularly aggressive phenotype of breast cancer and other correlated neoplasms (i.e. SMUP).

Response 7: Thank you for the suggestion. Carcinoma associated fibroblasts (CAFs) are one abundant stromal cell type in a breast tumor microenvironment. CAFs secrete different factors such as CXCL12 and EGF, driving cancer progression, migration, and metastasis. Our study focuses on the heterogeneous response of cancer cells in a gradient of these factors in the tumor microenvironment. We emphasize in the revised manuscript that CXCL12 and EGF are factors from the tumor microenvironment.

Reviewer 2 Report

This study investigated that oscillatory ERK Signaling and Morphology Determine Heterogeneity of Breast Cancer Cell Chemotaxis via MEK-ERK and p38-MAPK Signaling Pathways. The results indicated that therapeutic approaches to chemotaxis are inextricably linked to the interruption of oscillatory signals. If it can be published, it will bring great reference value to readers. However, there are many deficiencies and even errors in the writing of the paper. The author needs to carefully modify the narration, as follows. 

1 Why did the authors choose to target breast cancer cell rather than liver cancer cell etc.

2 Because of the previous appearance, the abbreviation of "Variational System Identification " in line 684 of the manuscript can be used directly.

3 There are many writing and formatting errors in the manuscript such as CO2 in the lines of 101 and 128 should be changed to CO2. Meanwhile, the abbreviation of milliliter should be mL. Please check and correct each one of them.

4 The novelty of this study should be stated in the introduction and discussion.

Author Response

Summary: This study investigated that oscillatory ERK Signaling and Morphology Determine Heterogeneity of Breast Cancer Cell Chemotaxis via MEK-ERK and p38-MAPK Signaling Pathways. The results indicated that therapeutic approaches to chemotaxis are inextricably linked to the interruption of oscillatory signals. If it can be published, it will bring great reference value to readers. However, there are many deficiencies and even errors in the writing of the paper. The author needs to carefully modify the narration, as follows.

Point 1: Why did the authors choose to target breast cancer cell rather than liver cancer cell etc.

Response 1: CXCL12-CXCR4 and EGF-EGFR signaling pathways are important drivers for metastasis in breast cancer. Previous studies found heterogeneous responses of cancer cells arising from pre-established cell states. However, processes underlying oscillatory signaling networks downstream of CXCR4 and EGFR signaling that regulate heterogeneous chemotactic responses of cancer cells remain incompletely understood. This becomes a critical barrier for development of effective anti-metastatic therapies in breast cancer as even a single cancer cell that evades therapy could potentially lead to metastatic disease.

Point 2: Because of the previous appearance, the abbreviation of "Variational System Identification " in line 684 of the manuscript can be used directly.

Response 2: We modified the manuscript and used VSI directly in line 711.

Point 3: There are many writing and formatting errors in the manuscript such as “CO2” in the lines of 101 and 128 should be changed to “CO2”. Meanwhile, the abbreviation of milliliter should be mL. Please check and correct each one of them.

Response 3: We modified the formatting errors for CO2. We indicate changes with a yellow highlight in lines of 115 and 142. All the abbreviation of milliliter are changed from ml to mL.

Point 4: The novelty of this study should be stated in the introduction and discussion.

Response 4: We appreciate this suggestion from Reviewer 2. Reviewer 1 also raised questions about novelty of the study. Please refer to our responses 1 and 2 to Reviewer 1.